# Influence of Healthy Brand and Diagnosticity of Brand Name on Subjective Ratings of High- and Low-Calorie Food

**DOI:** 10.3390/bs13010070

**Published:** 2023-01-13

**Authors:** Chengchen Zhang, Jiayi Han, Xiyu Guo, Jianping Huang

**Affiliations:** 1Department of Psychology, Soochow University, Suzhou 215123, China; 2School of Humanities and Social Science, The Chinese University of Hong Kong, Shenzhen 518172, China

**Keywords:** healthy brand, food calorie, diagnosticity of brand name, likeability, willingness to pay

## Abstract

Brand names on food packaging and the diagnosticity of brand names have notable effects on consumer preferences. However, their effects on healthy food consumption are not clear. The objective of this study was to investigate the effect of healthy brands and the diagnosticity of brand names on consumers’ subjective ratings of different calorie foods. In two studies, participants viewed 32 pictures of high- and low-calorie food product packaging from healthy and unhealthy brands and rated their feelings and willingness to pay online. Study 1 used real brand names, and Study 2 used fictional brand names and added press releases to manipulate diagnosticity. The present study demonstrated that participants perceived foods from healthy brands as healthier but less delicious and were more willing to buy low-calorie foods from healthy brands. Moreover, only when the brand name was of high diagnosticity were high-calorie foods rated as more likable, and did the willingness to pay for low-calorie foods increase. Collectively, these findings highlight the influence of the healthy brand on consumers’ subjective ratings of food. It is also inspiring for healthy food marketing.

## 1. Introduction

The healthy diet has become a global trend. To attract consumers, health labels are increasingly appearing on food packaging, such as ‘reduced in salt’ and ‘reduced fat’. Although health labels can convey health messaging, they may reduce consumers’ flavor expectations of food and inhibit their purchase intention [1]. The brand is tied to the general messaging of the product and can convey health messaging similar to health labels. For example, the fitness brand “Keep” tends to be perceived as healthy, while the chip brand “Lay’s” tends to be perceived as unhealthy. However, little research has been done on how the health messaging conveyed by brands affects consumers’ behavior. Do healthy brands influence consumers’ subjective ratings and purchase intention as health labels? Could this effect be affected by food calories? We believe that both high- and low-calorie foods from healthy brands are perceived as healthier but less delicious. Consumers may find high-calorie foods from healthy brands less delicious.

In addition, we propose that the diagnosticity of brand name also has an impact on food ratings. This study indicates that consumers perceive high-calorie foods as more likable and that they are more likely to purchase low-calorie foods only when the brand name is real (i.e., high diagnosticity). Previous studies have paid little attention to the effect of health messaging sent by healthy brands on consumer responses. Our study intends to fill this gap and provide implications for marketers to promote healthy food products.

### 1.1. The Effect of Health Labels and Brand Information on High- and Low-Calorie Foods

Food packaging is an important extrinsic cue influencing consumers’ perception or ratings of numerous different foods and drinks [2]. Food packaging usually includes many elements, such as food name, food picture, brand name, ingredient information, food labels, and so on. All of these elements can help set consumers’ expectations concerning the food product. For instance, health labels can convey health messaging and thus increase consumers’ expectations of product healthiness [3]. A strong negative correlation was found between perceived healthiness and calorie content [4]. When the picture of healthy food (e.g., vegetables) is presented on the packaging of an unhealthy product (e.g., mayonnaise), the perception of food calories becomes lower [5]. Specifically, individuals tend to associate high-calorie foods with low healthiness and low-calorie foods with high healthiness. Therefore, the perceived calorie content of the unhealthy product with healthy ingredients is lower than that of the unhealthy product without healthy ingredients or the unhealthy product with unhealthy ingredients.

The effect of health labels on consumption behavior is complicated. Some laboratory and field studies have shown that health labels on menus reduce consumers’ calorie intake [6,7]. However, Fenko and Faasen (2014) found that health labels on menus in healthy or unhealthy restaurants had no effect on consumers’ food choices [8]. Importantly, McCann et al. (2013) discovered that health labels, such as “low fat” or “low calorie”, may lead to more food consumption [9]. This health halo occurs when consumers determine food products’ overall healthfulness based on a limited number of attributes [10]. A positive health halo can lead to a “healthy = eat more” consumption pattern. Similarly, there is a heuristic named “unhealthy = delicious” during food choice. These findings reveal that stressing the healthiness of food through labeling may negatively affect consumers’ evaluation of deliciousness, convenience, etc. For example, the “low salt” label possibly reduces consumers’ expectations and actual taste ratings of foods [11].

In addition to health labels, brand information on food packaging is also a critical attribute influencing consumers’ taste perception and purchase intention. Consumers rely more on brand than taste to determine juice flavor [12]. Brand awareness, familiarity, popularity, and healthfulness can influence consumers’ food ratings. Juice from popular brands received higher food ratings versus that from unpopular brands [13]. Consumers also preferred the soy sauce flavor from a more familiar brand [14]. Moreover, healthy or unhealthy food brands can influence the health, calorie, and price ratings of foods [15]. Specifically, healthy brands would increase the perceived healthfulness and estimated price of unhealthy foods while decreasing the perceived caloric content. 

In general, both health labels and healthy brands can affect the subjective evaluation of food, such as deliciousness, healthiness, purchase intention, price estimation, etc. The effect of health labels on subjective evaluations has been demonstrated for many different types of food. However, the effect of health brands on subjective evaluation involves relatively few types of food. Moreover, in addition to the effect on subjective evaluation, health labels can even affect consumers’ intake of different calorie foods. 

Similar to health labels, healthy brands can also convey health messaging. Both health labels and brand information can affect high- and low-calorie food consumption. Consumers who focus on food calorie labels tend to choose lower-calorie foods in the Subway store [16]. Moreover, consumers believe that foods in Subway (i.e., healthy fast food) have fewer calories than foods in MacDonald’s (i.e., unhealthy fast food) [17]. Therefore, we propose that healthy brands influence consumers’ subjective ratings and purchase intention with respect to high- and low-calorie foods as well as health labels. The hypothesis is as follows:

**H1:** 
*For both high- and low-calorie foods, healthy brands increase participants*
*’ purchase intention while decreasing participants*
*’ subjective ratings.*


### 1.2. The Moderating Effect of the Diagnosticity of Brand Name

Diagnosticity refers to the contribution that the current information can make to solving the consumption problem faced by consumers [18]. The information is diagnostic if it helps consumers categorize a product, whereas it lacks diagnosticity when the information is ambiguous [19]. When consumers select products, information with a high level of diagnosticity is more likely to influence consumers’ evaluation of products [20]. The diagnosticity of the label could affect consumers’ food perception even if consumers did not consume or taste the food [21]. For example, consumers thought chocolate tastes stronger when it is labeled with “dark” rather than “milk” because “dark” sends diagnostic information related to the color of the chocolate [22].

According to the elaboration likelihood model, consumers may use relatively important attributes to evaluate products without sufficient cues [23]. For example, consumers may rely on origin, brand, and price to evaluate the quality of the product. However, when none of these cues is shown to consumers, they may rely on food packaging to make an evaluation. The diagnosticity of food labels has been proven to affect perceptions and choices of foods [24]. Therefore, the diagnosticity of brand name can also be used as an important cue to evaluate and select products. 

Notably, Yorkston and Menon (2004) found that only the true brand name, i.e., the brand name when a product comes to market, affected the taste expectation of food because true brand names possess high diagnosticity [25]. Conversely, the diagnosticity of the test name used for product testing purposes is low. Consumers are less likely to depend on test brand names to evaluate foods [26]. Therefore, we suggest that the diagnosticity of brand name can influence consumers’ evaluations of high- and low-calorie food products from healthy or unhealthy brands. The hypothesis is as follows:

**H2:** 
*For both high- and low-calorie foods, only the true healthy brand name, rather than the test healthy brand name, would increase participants’ purchase intention but decrease their subjective ratings.*


## 2. Study 1

The purpose of the present study was to investigate the influence of healthy brands on consumers’ subjective ratings and purchase intention with respect to different calorie food products.

### 2.1. Method

#### 2.1.1. Participants

A total of 156 Chinese adults (21.0 ± 2.09 years, ranging from 18 to 30 years, 39 males) were recruited. The present and the following experiments were approved by the ethics committee of the Psychology Department of Soochow University and performed following the ethical standards laid down in the Declaration of Helsinki. All participants gave informed consent electronically before the experiment started. We used the G*Power software, version 3.1.9.7 (Franz Faul, Christian-Albrechts-Universität Kiel, Kiel, Germany), to estimate the sample size and the results revealed that a sample of 54 participants can detect the effects with η_p_^2^ = 0.25 (statistical power = 0.95).

#### 2.1.2. Materials

To select foods and brands in the formal experiment, similar to Masterson et al. (2020) [15], we conducted two pretests. We chose 10 high-calorie (e.g., cucumber, carrots, tomatoes, etc.) and 10 low-calorie (e.g., popcorn, cookies, sausage, etc.) foods based on the calorie ranges. Each original food packaging picture contained only the food name and food picture. We also chose 3 healthy and 3 unhealthy brands. Of the participants, 95 estimated the calorie content and familiarity of 20 foods on a 7-point Likert scale. Further, 60 participants were asked to rate the healthiness, familiarity, and willingness to pay for 6 brands. All participants lived in China and were between 18 and 52 years of age.

The 8 foods with the highest calorie rating (*M* = 5.72, *SD* = 1.12) and the 8 foods with the lowest calorie ratings (*M* = 2.78, *SD* = 1.48) were categorized as high- and low-calorie foods, respectively (*t* (94) = 32.63, *p* < 0.001, Cohen’s *d* = 1.49, 95% CI = [2.76, 3.11]). Similarly, the brand “Keep” with the highest healthfulness ratings (*M* = 5.23, *SD* = 1.98) and the brand “Lay’s” (*M* = 4.80, *SD* = 2.01) with the lowest healthfulness ratings were categorized as healthy and unhealthy brands, respectively (*t* (94) = 10.43, *p* < 0.001, Cohen’s *d* = 0.19, 95% CI = [1.94, 2.84]). Their familiarity and willingness to pay were not significantly different (all *p*s > 0.05). 

The formal experiment was conducted online at www.qualtrics.com (accessed on 17 December 2022). We used Adobe Photoshop CS6 to add brand names to original food packaging pictures. Finally, 32 food packaging pictures (694 pixels wide × 982 pixels high) of the healthy brand, Keep, and the unhealthy brand, Lay’s, were shown to the participants (see Appendix A). Each brand had 8 high-calorie and 8 low-calorie food products. The food packaging only contained brand name, food name, and food picture.

#### 2.1.3. Design and Procedure

A 2 (brand type: healthy brand, Keep; unhealthy brand, Lay’s) × 2 (food calories: high; low) within-subject design was used. The pictures were presented in a random order, with one picture on each page. Similar to the PERVAL evaluation scale [26], during each trial, the participants were shown one picture and asked to rate deliciousness, healthiness, likability, pleasantness, and willingness to buy this food on 7-point scales. They were also asked to indicate their willingness to pay for each food product by specifying the amount of money (1–20 ¥). Notably, willingness to buy indicates the likelihood of purchase behavior [27], while willingness to pay indicates the amount participants who are willing to pay for the product [28]. At the end of the survey, the healthiness and familiarity of Keep and Lay’s were rated on a 7-point scale. 

### 2.2. Results and Discussion

#### 2.2.1. Brand Ratings

In the present and the following studies, we used SPSS 22 software (Statistical Product and Service Solutions, IBM) to conduct all the data analyses. Based on the *t*-test results, the Keep brand (*M* = 5.17, *SD* = 1.49) was significantly healthier than the Lay’s brand (*M* = 3.49, *SD* = 1.39) (*t* (155) = 8.21, *p* < 0.001, Cohen’s *d* = 0.66, 95% CI = [0.85, 1.38]). Lay’s (*M* = 5.29, *SD* = 1.41) was more familiar than Keep (*M* = 4.17, *SD* = 1.82) (*t* (155) = 10.47, *p* < 0.001, Cohen’s *d* = 0.84, 95% CI = [1.36, 2.00]).

These results revealed that Keep is rated as healthier, which is in line with the pretest result. In contrast to the pretest result, however, Lay’s was rated as more familiar. This discrepancy can be explained by the difference in participant groups.

#### 2.2.2. Subjective Ratings

The mean scores of the deliciousness, healthiness, likability, and pleasantness ratings are shown in Figure 1. We first performed a 2 (brand type: healthy brand; unhealthy brand) × 2 (food calories: high; low) repeated-measures ANOVA (see Table 1 for a summary of the results). The results revealed the significant main effects of brand and food calories on all four measures (see Figure 2). The food products from the unhealthy brand were perceived as more delicious (4.32 vs. 4.08), likable (4.3 vs. 4.2), and pleasant (4.3 vs. 4.19), but less healthy (3.79 vs. 4.27) than those from the healthy brand. High-calorie food products were rated as more delicious (4.42 vs. 3.98), likable (4.38 vs. 4.12), and pleasant (4.43 vs. 4.06), but less healthy (4.42 vs. 4.77) than low-calorie food products. As shown in Table 1, the results also revealed a significant interaction effect between brand type and food calories on three measures except for healthiness.

In order to interpret the significant Brand type × Food calories interaction terms on three measures, a one-way repeated-measure ANOVA was performed for each type of food calorie content with brand type as the independent factor. The results revealed that the high- and low-calorie foods from Lay’s were rated as more delicious than those from Keep (*M*_H_: 4.58 vs. 4.26; *M*_L_: 4.06 vs. 3.90) (*F*_H_ (1, 155) = 32.69, *p*_H_ < 0.001, η_p_^2^_H_ = 0.174; *F*_L_ (1, 155) = 10.42, *p*_L_ = 0.002, η_p_^2^_L_ = 0.063). Moreover, the high-calorie foods from Lay’s had significantly higher likability (4.48 vs. 4.27) and pleasantness (4.54 vs. 4.31) ratings than those from Keep (*F_L_* (1, 155) = 16.41, *p_L_* < 0.001, η_p_^2^_L_ = 0.096; *F_P_* (1, 155) = 18.29, *p_P_* < 0.001, η_p_^2^_P_ = 0.106). No such difference was observed for the low-calorie foods (all *p*s > 0.05). 

In summary, these results revealed that high-calorie foods and unhealthy brand foods were rated as more delicious, likable, and pleasant, but unhealthier. Moreover, participants had higher pleasantness and likability ratings of high-calorie foods from the unhealthy brand. The combination of the unhealthy brand and low-calorie foods may create a cognitive conflict and therefore reduce pleasantness and likability ratings. Similarly, health labels on unhealthy products (e.g., potato chips) can also negatively affect evaluations [29].

#### 2.2.3. Purchase Intention

The mean scores of willingness to buy and willingness to pay are also shown in Figure 1. As shown in Table 1, the main effects of brand type and food calories on willingness to buy and willingness to pay were not significant (all *p*s > 0.05). However, a significant interaction effect between brand type and food calories on willingness to buy was found.

In order to interpret the significant interaction terms on willingness to buy, the data from the two groups of participants were combined and a one-way repeated-measure ANOVA was performed for each type of food calorie with brand type as the independent factor. The results revealed that participants had a significantly higher willingness to buy high-calorie foods from Lay’s than from Keep (3.98 vs. 3.83) (*F* (1, 155) = 5.93, *p* = 0.016, η_p_^2^ = 0.037). Moreover, participants had a significantly higher willingness to buy low-calorie foods from Keep than from Lay’s (3.85 vs. 3.73) (*F* (1, 155) = 4.54, *p* = 0.035, η_p_^2^ = 0.028).

In summary, participants showed a tendency to buy low-calorie food products from the healthy brand and high-calorie food products from the unhealthy brand. Notably, the willingness to pay results showed no significant difference. Participants’ brand familiarity and price perceptions for Keep and Lay’s products may affect their willingness to pay. In addition, participants’ health motivations, taste preferences, and appetites may play a role. Therefore, in Study 2, we used fictional brands to control the effect of brand familiarity and added the diagnosticity of brand name to investigate whether it affects the influence of healthy brands on subjective ratings and purchase intention.

## 3. Study 2

The purpose of the present study was twofold. First, we investigated the influence of fictional healthy brands on consumers’ subjective ratings and purchase intention with respect to different calorie food products. Second, we investigated how the diagnosticity of brand name influences the effect of healthy brands consumers’ subjective ratings and purchase intention.

### 3.1. Method

#### 3.1.1. Participants

A total of 146 new Chinese adults (21.8 ± 2.38 years, ranging from 19 to 30 years, 30 males) were recruited. They were divided into the true brand group (i.e., high diagnosticity) (*N* = 73, 21.3 ± 1.77 years, 11 males) and the test brand group (i.e., low diagnosticity) (*N* = 73, 22.2 ± 2.81 years, 19 males). We used the G*Power software to estimate the sample size, and the results revealed that a sample of 36 participants can detect the effects with η_p_^2^ = 0.25 (statistical power = 0.95).

#### 3.1.2. Materials, Design and Procedure

All aspects were the same as those of Study 1 except for the following differences. First, 32 food packaging pictures (694 pixels wide × 982 pixels high) of the fictional brands “Nish” or “Cerp” were shown to the participants (see Appendix B). Each brand had 8 high-calorie and 8 low-calorie food stimuli. The original food pictures were the same as in Study 1. Based on the pretest results (*N* = 95), no significant difference in participants’ healthiness and familiarity with Nish and Cerp was found (all *p*s > 0.05). 

Second, a mixed design of 2 (brand type: healthy brand; unhealthy brand) × 2 (food calories: high; low) × 2 (diagnosticity of brand name: true name; test name) was used in Study 2. The brand type and food calories were the within-group independent variables, and diagnosticity of brand name was the between-group independent variable.

Third, in order to manipulate diagnosticity, we used two press releases about the launch of the new products from the new brands “Cerp” or “Nish”. One sent health messaging, while the other conveyed neutral information (see Appendix C). Participants first saw one release and evaluated 16 high- and low-calorie food pictures of the brand mentioned in the release. After that, they read another release and evaluated the remaining 16 pictures. At the end of the survey, participants were asked to complete the brand evaluation.

### 3.2. Results and Discussion

#### 3.2.1. Brand Ratings

Based on the *t*-test results, participants’ familiarity with Cerp and Nish showed no significant difference (*p* > 0.05). Participants rated the healthiness of the healthy brand (*M* = 4.35, *SD* = 1.43) significantly higher than that of the unhealthy brand (*M* = 4.01, *SD* = 1.42) (*t* (145) = 2.97, *p* = 0.004, Cohen’s *d* = 0.25, 95% CI = [0.11, 0.55]).

A significant difference between the two brands was found in healthiness but not in familiarity. This suggested that brand familiarity did not affect product evaluation and that press releases manipulated the brand type successfully.

#### 3.2.2. Subjective Ratings

The mean scores of subjective ratings are shown in Figure 3. We first performed a 2 (brand type: healthy brand; unhealthy brand) × 2 (food calories: high; low) × 2 (diagnosticity of brand name: true name; test name) mixed ANOVA. The results are summarized in Table 2. 

The results revealed a significant main effect of brand type on all measures except deliciousness. The foods from the unhealthy brand were perceived as more likable (4.29 vs. 4.21), pleasant (4.26 vs. 4.18), and less healthy (4.10 vs. 4.23) than those from the healthy brand (see Figure 4). The results also revealed a significant main effect of food calories on all four measures. The high-calorie food products were perceived as more delicious (4.46 vs. 4.06), likable (4.37 vs. 4.13) and pleasant (4.38 vs. 4.05), but less healthy (3.48 vs. 4.84) than the low-calorie food products (see Figure 4).

We also found a significant interaction between the diagnosticity of brand name × food calories on likability. Further analysis showed that in the true name group, the high-calorie foods were rated as more likable than low-calorie foods (4.44 vs. 4.05) (*F* (1, 144) = 18.60, *p* < 0.001, η_p_^2^ = 0.114). Such an effect was not found in the test name group (*p* > 0.05). None of the other main or interaction effects were significant.

In summary, the subjective rating results were consistent with that of Study 1, except that the main effect of brand type on deliciousness was not significant. It is difficult for consumers to imagine the flavor based on the fictional brand name. Compared with Lay’s and Keep, the food products from Cerp and Nish brand are strange, weird, and unfamiliar.

Notably, participants liked high-calorie foods more than low-calorie foods only in the true brand name group. No such difference was found in the test name group. This revealed that participants were unable to depend on the health messaging conveyed by the test brand name to evaluate the product. Low diagnosticity information can hardly be used as cues to make an evaluation [20].

#### 3.2.3. Purchase Intention

The mean scores of willingness to buy and willingness to pay are also shown in Figure 3. As shown in Table 2, the main effects of brand type and food calories on willingness to pay were significant. The willingness to pay scores of the healthy brand and low-calorie foods were significantly higher than those of the unhealthy brand and high-calorie foods (M_B_: 6.72 vs. 6.52; M_F_: 6.82 vs. 6.36).

The results also revealed a significant triple interaction among the three factors on the willingness to pay scores (*F* (1, 144) = 3.91, *p* = 0.050, η_p_^2^ = 0.026). Further analysis showed a significant main effect of food calories on willingness to pay only in the true brand name group (*F* (1, 72) = 9.37, *p* = 0.003, η_p_^2^ = 0.115). Specifically, compared to the high-calorie foods, participants were willing to pay more for the low-calorie foods in the true brand name group (6.60 vs. 7.18). None of the other main or interaction effects were significant.

In summary, participants were willing to pay more for low-calorie foods from the healthy brand. Similar to organic foods, consumers would like to pay more for the health value of healthy brand foods [9]. Moreover, participants were willing to pay more for low-calorie foods only in the true brand name group. One possible reason is that the situation is closer to the real purchasing situation in the true name group, in which consumers are more cautious about paying for different calorie products.

## 4. General Discussion

Taken together, three major findings emerged from the present study. First, consumers perceived both high- and low-calorie foods from healthy brands to be healthier but less delicious than those from unhealthy brands. However, consumers considered high-calorie foods from the healthy brand to be less pleasant and likable (as in Study 1). These results are in line with the “unhealthy = delicious” heuristic [30]. Collectively, these results demonstrated that the healthy brand could decrease participants’ subjective ratings of different calorie food products, which partly confirmed H1. 

Second, participants only had a significantly higher willingness to buy low-calorie foods from the healthy brand (as in Study 1). One possible explanation is the expectancy violation effect. Healthy or unhealthy brands create the expectation of a healthy or unhealthy food image. However, high-calorie (low-calorie) foods from the healthy (unhealthy) brand violate consumers’ expectations. Therefore, the willingness to buy the high(/low)-calorie foods from the healthy(/unhealthy) brand was negatively affected [31]. 

Third, consumers perceived high-calorie foods as more likable, and their willingness to pay for low-calorie foods was increased only in the true brand name group. No such findings existed in the test brand name group (as in Study 2). These results implied that the diagnosticity of a brand name can influence food ratings and purchase intention, which partly confirmed H2. Therefore, the diagnosticity of a brand can play an important role in food choice. A recent study also proved that the diagnosticity of brands affected hotel booking intention [32].

### 4.1. Theoretical and Practical Implications

Previous studies have focused on the effect of price [32], origin [33], and color [34] of food packaging on consumption behavior while ignoring the brand. Health labels’ effects on food ratings in healthy consumption have also received much attention [35,36,37,38]. In addition, the brand on food packaging plays an important role in food ratings. As brands are tied to the general messaging and products of their product line [39], a brand selling mostly healthy products tends to have a healthy image. Thus, the brand is more likely to be perceived as a healthy brand if it sells mostly healthy products. Previous research has proven that healthy or unhealthy food brands can influence the health, calorie, and price ratings of foods [15]. Specifically, healthy brands would increase the perceived healthfulness and estimated price of unhealthy foods while decreasing the perceived caloric content. However, researchers have paid little attention to how the health messaging conveyed by healthy brands on packaging affects consumers’ subjective ratings of different calorie food products. Our research proved that healthy brands on packaging affected consumers’ subjective ratings of different calorie food products, and thus contributed to filling this gap.

In addition, this research also investigated how the diagnosticity of brand names changed participants’ subjective ratings and purchase intentions. Previous studies have demonstrated that only the true brand name (with high diagnosticity) affects the taste expectation of food [25]. Although the main effect of the diagnosticity of a brand name in our study was not significant, we proved that the diagnosticity of a brand name can influence the subjective ratings of different calorie food products. Consumers perceived high-calorie foods as more likable, and their willingness to pay for low-calorie foods was increased only in the true brand name group (with high diagnosticity). Recent research has also demonstrated that high-diagnosticity food labels can improve perceptions and influence food choice [24]. Thus, high diagnosticity is an important factor in improving food ratings and purchase intentions.

For manufacturers, since consumers perceive foods with healthy brands as healthier but less delicious, they should promote the deliciousness of healthy brand foods. Importantly, manufacturers should be careful to launch products inconsistent with the brand’s previous image; otherwise, it may reduce consumers’ food evaluations. In addition, in order to increase purchase intention, manufacturers can improve the diagnosticity of brands by using colors that represent healthy food or by adding attributes conveying health messaging.

### 4.2. Limitations and Future Research

First, each participant needed to evaluate 32 pictures, which may cause a fatigue effect and affect the results. Second, in Study 2, we only conveyed the diagnosticity of brand name to participants through a simple guideline, which may not serve the purpose effectively. Third, we presented the products to participants in the form of pictures, which is quite different from the shopping situation in stores, leading to low ecological validity. Fourth, there were still some problems with participant recruitment in our study. The sample diversity was insufficient, and most of the participants were college students. The quantity of participants was also a little bit small. Most importantly, gender was likely to modulate the subjective ratings. The quantity of male participants is relatively small in Study 2; only 30 of the 146 participants were male. Therefore, the results were biased by the female participants.

Follow-up studies can reduce the usage of food picture stimuli and select longer and more accurate press releases and instructions to ensure participants’ reading time. Moreover, food preference could affect consumer choice. The health motivations, taste preferences, and appetites of the participants should be taken into consideration as covariates in study design. VR (virtual reality) technology could be adopted to improve the external validity of the study. Finally, recruiting different types of consumers and more male participants in future studies could enhance the generalizability of the results. For example, a cross-cultural study could be conducted to investigate the effect of health brands and the diagnosticity of brand names on the food ratings of consumers from different countries.

## Figures and Tables

**Figure 1 behavsci-13-00070-f001:**
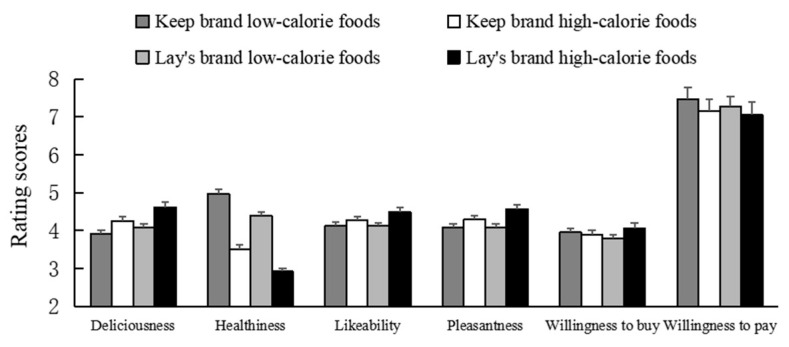
The mean scores of subjective ratings and purchase intention in Study 1. Error bars show the standard errors of the means.

**Figure 2 behavsci-13-00070-f002:**
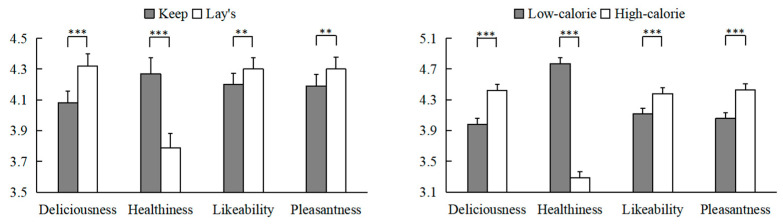
The main effects of brand type and food calories on subjective ratings and purchase intention in Study 1. Error bars show the standard errors of the means. ** denotes *p* < 0.01 and *** denotes *p* < 0.001.

**Figure 3 behavsci-13-00070-f003:**
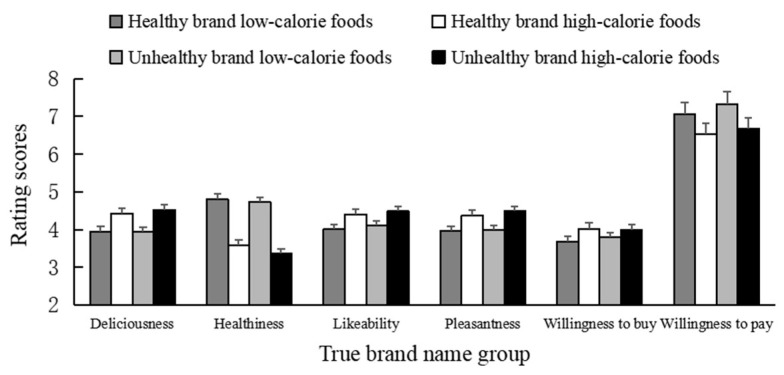
The mean scores of subjective ratings and purchase intention in Study 2. Error bars show the standard errors of the means.

**Figure 4 behavsci-13-00070-f004:**
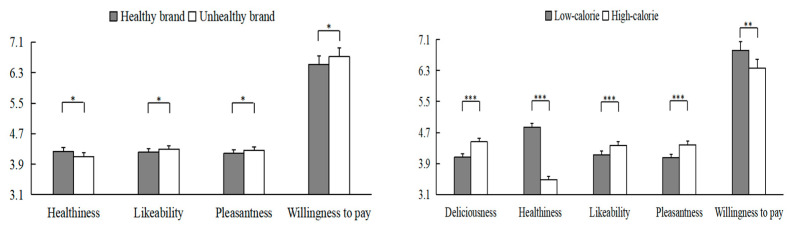
The main effects of brand type and food calories on product evaluation in Study 2. Error bars show the standard errors of the means. * denotes *p* < 0.05, ** denotes *p* < 0.01, and *** denotes *p* < 0.001.

**Table 1 behavsci-13-00070-t001:** The results of ANOVAs conducted on the data collected in Study 1.

Effect	Measure	*F* (1, 155)	η_p_^2^	Measure	*F* (1, 155)	η_p_^2^
Brand type	Deliciousness	28.19 ***	0.15	Willingness to buy	0.09	-
	Healthiness	60.71 ***	0.28	Willingness to pay	0.95	-
	Likeability	8.41 **	0.05			
	Pleasantness	8.19 **	0.05			
Food calories	Deliciousness	52.83 ***	0.25	Willingness to buy	2.62	-
	Healthiness	377.78 ***	0.71	Willingness to pay	3.49	-
	Likeability	20.02 ***	0.11			
	Pleasantness	35.35 ***	0.19			
Brand type	Deliciousness	9.63 **	0.06	Willingness to buy	13.55 ***	0.08
×	Healthiness	0.33	-	Willingness to pay	0.44	-
Food calories	Likeability	10.13 **	0.06			
	Pleasantness	15.70 ***	0.09			

Note: ** *p* < 0.01, and *** *p* < 0.001.

**Table 2 behavsci-13-00070-t002:** The results of ANOVAs conducted on the data collected in Study 2.

Effect	Measure	*F* (1, 144)	η_p_2	Measure	*F* (1, 144)	η_p_2
Brand type	Deliciousness	0.78	-	Willingness to buy	0.59	-
	Healthiness	5.95 *	0.04	Willingness to pay	6.03 *	0.04
	Likeability	4.75 *	0.03			
	Pleasantness	4.03 *	0.03			
Food calories	Deliciousness	33.44 ***	0.19	Willingness to buy	2.43	-
	Healthiness	177.70 ***	0.55	Willingness to pay	11.15 **	0.07
	Likeability	15.16 ***	0.10			
	Pleasantness	28.37 ***	0.17			
Diagnosticity	Deliciousness	0.36	-	Willingness to buy	0.85	-
	Healthiness	0.40	-	Willingness to pay	1.64	-
	Likeability	0.00	-			
	Pleasantness	0.02	-			
Brand type	Deliciousness	6.30 *	0.04	Willingness to buy	0.14	-
×	Healthiness	5.00 *	0.03	Willingness to pay	0.85	-
Food calories	Likeability	0.00	-			
	Pleasantness	2.75	-			
Brand type	Deliciousness	0.03	-	Willingness to buy	0.10	-
×	Healthiness	0.12	-	Willingness to pay	0.00	-
Diagnosticity	Likeability	0.00	-			
	Pleasantness	0.05	-			
Food calories	Deliciousness	3.68	-	Willingness to buy	3.79	-
×	Healthiness	0.42	-	Willingness to pay	2.04	-
Diagnosticity	Likeability	4.86 *	0.03			
	Pleasantness	3.47	-			
Brand type	Deliciousness	0.68	-	Willingness to buy	2.66	-
×	Healthiness	0.25	-	Willingness to pay	3.91 *	0.03
Food calories	Likeability	0.10	-			
×	Pleasantness	0.16	-			
Diagnosticity						

Note: * *p* < 0.05, ** *p* < 0.01, and *** *p* < 0.001.

## Data Availability

Data is contained within the article and can be made available upon request.

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
