# Peer review of "Influence of Healthy Brand and Diagnosticity of Brand Name on Subjective Ratings of High- and Low-Calorie Food"

_behavsci, 2023, doi:10.3390/bs13010070_

Round 1

Reviewer 1 Report

The article deals with an interesting topic the effects of brand name on consumer behavior. But it has a few major limitations.

(1)   Line 41-42: Previous researches paid little attention to the effect of health messaging sent by healthy brands on consumer behavior. However, there are lots of researches about the effect of health messaging sent by healthy labels on consumer behavior. The authors should discussed the different effects of brands and labels in more detail.

(2)   Line 52-54: When the picture of healthy food....., the perception of food calories became lower. The authors should provide more information to support this conclusion.

(3)   Line 122-123: 32 food packaging pictures (694 pixels wide × 982 pixels high) of healthy brand Keep and unhealthy brand Lays were shown to the participants. The authors should explain why brand Lays were unhealthy although the results of a pre-experiment showed Keep was healthier than Lay's.

(4)   Line 125-126: The food packaging only contains brand name, food name, and 125 food picture, without any other labels. How to compare the food calories of different brands as no information on calories was present.

(5)   Line 114-120: Food prefer could affect cconsumer choice. Thus, this paper should exclude the effect of consumers' background knowledge (health motivation, taste preference, and appetite) in the experiment.

(6)   Line 341-350: Contrary to the previous study....formation with low diagnosticity might be ignored. The author should reconsider the title of this paper.

(7)   Line 357-359: This research investigated how...which fulfilled this gap The authors should explain what is healthy brands in more detail.

Author Response

Reviewer(s)' Comments to Author:

Reviewer: 1

Comments and suggestions: The article deals with an interesting topic the effects of brand name on consumer behavior. But it has a few major limitations.

Thank you very much for your constructive and detailed comments and suggestions regarding our manuscript. We have revised the manuscript based on your comments and suggestions, as follows.

Additional Questions:

  • Line 41-42:Previous researches paid little attention to the effect of health messaging sent by healthy brands on consumer behavior. However, there are lots of researches about the effect of health messaging sent by healthy labels on consumer behavior. The authors should discussed the different effects of brands and labels in more detail.

In the revised version of the manuscript, we have discussed the effects of brands and labels on consumer behavior in more detail in the introduction section (see line 61-89 in page 2). We unraveled the complexity of the effect of health labels and healthy brands on consumer perception, purchase intention as well as eating behavior, and summarized the different effects of healthy labels and brands. This could provide the reader with more comprehensive background information

(2) Line 52-54:When the picture of healthy food....., the perception of food calories became lower. The authors should provide more information to support this conclusion.

In the revised version of the manuscript, we have provided more information to support this conclusion (see line 54-60 in page 2). Because the picture of healthy food is a healthy ingredient, it has a strong negative correlation with perceived calorie content. If it appears on an unhealthy product, the perceived calorie content becomes lower.

(3) Line 122-123:32 food packaging pictures (694 pixels wide × 982 pixels high) of healthy brand Keep and unhealthy brand Lays were shown to the participants. The authors should explain why brand Lays were unhealthy although the results of a pre-experiment showed Keep was healthier than Lay's.

In the revised version of the manuscript, we have rewritten the material selection process in the section of Materials (see line 145-167 in page 4). Our selection process of Lay’s as an unhealthy brand was in reference to the study by Masterson et al.(2020).  Participants evaluated the healthfulness of 3 healthy and 3 unhealthy brands in the pretest. Keep got the highest healthfulness ratings and Lay’s got the lowest healthfulness ratings. Thus they were defined as the healthy brand and the unhealthy brand respectively.

(4) Line 125-126:The food packaging only contains brand name, food name, and food picture, without any other labels. How to compare the food calories of different brands as no information on calories was present.

In the revised version of the manuscript, we have rewritten the material selection process in the section of Materials (see line 145-167 in page 4). The original high- and low-calorie food packaging pictures we chose were based on high-calorie food and low-calorie food calorie range in Toepel et al.(2009)’s study., only contained food names and food pictures. Moreover, in the pre-experiment, the original high-calorie food pictures had significantly higher calorie ratings than the low-calorie food pictures (M = 2.78, SD = 1.48). Therefore, in the formal experiment, regardless of the brand the food came from, when the food picture was presented, consumers could identify it as a high or low-calorie food.

(5) Line 114-120: Food prefer could affect consumer choice. Thus, this paper should exclude the effect of consumers' background knowledge (health motivation, taste preference, and appetite) in the experiment.

I'm sorry that we didn't take consumers' health motivation, taste preference and appetite into consideration in these two studies. We acknowledge that this is a limitation of our study and discussed this issue in the section 4.2 Limitations and future research (see line 438-439 in page 11) and planned to recruit different types of participants in the future study. We admit that taste preference will affect consumers' evaluation. But all participants were shown the same pictures of food from different brands. Thus, the taste preference could be balanced.

(6) Line 341-350:Contrary to the previous study....formation with low diagnosticity might be ignored. The author should reconsider the title of this paper.

In the revised version of the manuscript, our title has been changed as‘Influence of healthy brand and diagnosticity of brand name on subjective ratings of high- and low-calorie food’(see line 2-3 in page 1). 

(7) Line 357-359:This research investigated how...which fulfilled this gap The authors should explain what is healthy brands in more detail.

In the revised version of the manuscript, we have explained what is healthy brands in more detail (see line 396-403 in page 11).

Reviewer 2 Report

The presented study is quite interesting. However there are some minor flaws I would like to adress:

1. The quantity of participants in second study is too small.

2. Abbreviations used in any place (tables, figures) must be expanded. Tables as well as figures suppose to be self-read. 

3. English requires a slight correction.

4. The discussion need to be deepened.

5. There is no data on food packaging design in the manuscript but the authors made a conclusion on it in the abstract (see line 19). It should be reformulated.

Author Response

Reviewer: 2

Comments and suggestions : The presented study is quite interesting. However there are some minor flaws I would like to adress:

Thank you very much for your constructive and detailed comments and suggestions regarding our manuscript. We have revised the manuscript based on your comments and suggestions, as follows.

Additional Questions:

  1. The quantity of participants in second study is too small.

In the revised version of the manuscript, we have corrected the sample size estimation in section Participants (see line 142-143 and 269-270). 36 participants can detect the effects with η2p = 0.25 (statistical power = 0.95). Therefore, the current sample size has been able to obtain a medium-size effect. We acknowledged that the quantity of participants in second study is too small, and added discussion of this issue in Limitations and future research section (see line 434-435 and 441-444 in page 11).

  1. Abbreviations used in any place (tables, figures) must be expanded. Tables as well as figures suppose to be self-read.

In the revised version of the manuscript, we have expanded abbreviations in tables and figures and changed all abbreviations in all sections. Tables as well as figures now, are self-read.

  1. English requires a slight correction.

We are sorry that there are still some problems with our manuscript, so we have carefully checked and requested the paper polishing company to revise our manuscript, including some grammatical issues, and so on. Specific modifications have been be marked up using the“Track Changes”function in the main text.

  1. The discussion need to be deepened.

In the revised version of the manuscript, we deepened the discussion in the section theoretical and practical implications. We discussed the effect of healthy brands and the modulating effect of diagnosticity of brand name in more aspects (see line 396-419 in page 11). On the one hand, this study paied attention to the health messaging conveyed by brands which was neglected in previous studies, and confirmed its effect on food ratings. On the other hand, this study also explored the more impact of the diagnosticity of brand name in the subjective evaluation of food.

  1. There is no data on food packaging design in the manuscript but the authors made a conclusion on it in the abstract (see line 19). It should be reformulated.

In the revised version of the manuscript, we have removed the food packaging design in the abstract (see lines 20 in page 1).

Round 2

Reviewer 1 Report

Moderate English changes required

Author Response

Reviewer(s)' Comments to Author:

Reviewer: 1

Thank you very much for your constructive and detailed comments and suggestions regarding our manuscript. We have revised the manuscript based on your comments and suggestions, as follows.

Additional Questions:

  1. Moderate English changes required

We are sorry that there are still some problems with our manuscript. So we have carefully checked and invited the native English editor to revise our manuscript, including some logical, grammatical, vocabulary usage issues and so on. Specific modifications have been be marked up using the “Track Changes” function in the main text.